# Measuring and Predicting the Effects of Residual Stresses from Full-Field Data in Laser-Directed Energy Deposition

**DOI:** 10.3390/ma16041444

**Published:** 2023-02-08

**Authors:** Efstratios Polyzos, Hendrik Pulju, Peter Mäckel, Michael Hinderdael, Julien Ertveldt, Danny Van Hemelrijck, Lincy Pyl

**Affiliations:** 1Department of Mechanics of Materials and Constructions, Vrije Universiteit Brussel (VUB), Pleinlaan 2, BE-1050 Brussels, Belgium; 2isi-sys GmbH, Wasserweg 8, D-34131 Kassel, Germany; 3Department of Mechanical Engineering, Vrije Universiteit Brussel (VUB), Pleinlaan 2, BE-1050 Brussels, Belgium

**Keywords:** 3D micro-DIC, incremental hole drilling, L-DED AISI 316L stainless steel, thermal expansion coefficient, residual thermal stresses, stochastic finite element modeling, supervised machine learning, polynomial chaos expansion

## Abstract

This article presents a novel approach for assessing the effects of residual stresses in laser-directed energy deposition (L-DED). The approach focuses on exploiting the potential of rapidly growing tools such as machine learning and polynomial chaos expansion for handling full-field data for measurements and predictions. In particular, the thermal expansion coefficient of thin-wall L-DED steel specimens is measured and then used to predict the displacement fields around the drilling hole in incremental hole-drilling tests. The incremental hole-drilling test is performed on cubic L-DED steel specimens and the displacement fields are visualized using a 3D micro-digital image correlation setup. A good agreement is achieved between predictions and experimental measurements.

## 1. Introduction

Laser applications have become increasingly prominent in recent years, with applications being found in a variety of fields, such as additive manufacturing [1].

Laser-directed energy deposition (L-DED) is an additive manufacturing process in which a part is created by simultaneously delivering both material and energy [2]. The material is typically introduced in a powder form, and it is melted using a laser beam that follows a specific pattern [3]. Once melted, the material is solidified almost instantaneously due to conduction and forms a solid layer. Compact parts are produced by the successive layer-by-layer deposition of melted powder on the surface of already solidified material [4,5].

The above process leads to parts that require minimal tooling and relatively little secondary processing. However, the manufacturing of parts using the L-DED is accompanied by residual thermal stresses [6]. Combined with the distortions, the residual stresses are inherent attributes of the thermal strains in the layer-by-layer printing process and are responsible for altering the mechanical behavior of manufactured parts [7]. In particular, they aid in deteriorating the fracture resistance of additively manufactured parts and especially their fatigue life [5]. Due to their pronounced effects, understanding the residual stresses is a necessity.

A number of works focus on distortion and residual stress characterization. Mukherjee et al. [8] performed meritorious simulation work, transferring the nodes, elements, and temperature data from 3D transient heat transfer and fluid flow models to an ABAQUS finite element (FE)-based mechanical model. The effect of the layer thickness on the residual stresses was studied.

Three-dimensional thermo-mechanical models at different scales (single-walled structures [9,10], rectangular and S-shaped structures [11], and part scale [12]) showed good agreement with the experimental results and were further explored to analyze the sensitivity of the process settings on distortions and residual stresses.

Bartlett et al. [13] converted heterogeneous surface curvatures to in-plane residual stress and demonstrated this on 316L stainless steel selective laser melting manufactured inverted-cone parts.

Similar studies were performed in multi-layer wire arc additive manufacturing using thermo-mechanical models. In the study of Wu et al. [14], the residual stresses were simulated by applying temperature-dependent material properties on elements activated as the heat source is moving, using an element birth technique. The quantification of residual stress/strain measurements relied on ex situ testing via incremental hole drilling (IHD). The authors measured the residual stresses in two locations on the specimens’ surface and compared the experimental results with the ones of the thermo-mechanical models.

However, measuring the effects of the residual stresses should not be limited in specific locations because local phenomena can affect the accuracy of experiments. Especially for L-DED manufactured parts, surface imperfections due to the local heat transfer by the laser beam are common. These imperfections can introduce noise and compromise the measurements.

An answer to this problem can be given by the use of full-field techniques, such as the digital image correlation (DIC). The DIC technique was introduced in 2008 [15] for providing the full-field displacement data on the surface of specimens tested in the IHD. Apart from capturing local imperfections, another main advantage of the full-field displacement data is providing information close to the drilling hole. Information like this is impossible to be derived using strain-measuring means, such as strain gauges. However, even when full-field data are obtained, their processing and use can be challenging due to the lack of information regarding the tools available for handling such data.

In summary, residual thermal stresses in additively manufactured parts significantly influence the behavior of the parts and require quantification [2]. Properly characterizing the effects of residual stresses is not trivial due to the surface roughness of the parts, and full-field methods must be implemented. Despite the importance of full-field data though, handling them appropriately is challenging. Therefore, studies that present novel approaches for tackling such complex problems and offering solutions using contemporary means are a necessity.

This study illustrates a novel approach for quantifying the effects of residual stresses for L-DED AISI 316L stainless steel parts. The approach focuses on leveraging the full-field data coming from DIC images to improve the accuracy of measurements and predictions alike. To this end, tools that exhibit rapidly growing potential such as machine learning (ML) and polynomial chaos expansion (PCE) are utilized in combination with numerical FE models. Both for the measurement and the prediction part of this study, ML- and PCE-based metamodels are used to provide a link between the experiments and the FE models, thus allowing to fully exploit the potential of full-field data.

The ML- and PCE-based metamodels are simplified representations of the underlying physics and are able to accurately predict the behavior of a system while requiring significantly less computational resources than a full FE model. This makes metamodels an ideal choice for optimization schemes that require iterative solutions of the model. This study explores the use of metamodels specifically for predicting the displacement field in a sample due to thermal expansion. The use of metamodels is further explained in Section 4.

## 2. Aim and Structure of the Approach

The aim of this approach is to use ML and PCE, two of the most decorated data processing techniques, to integrate experimental full-field data with numerical FE models. This approach can be separated into two parts, one associated with measurements and one with predictions. Even though both parts refer to the quantification of the effects of residual stresses, they present inherent differences that can be summarized as follows.

The part associated with measurements is focused on extracting material properties of thin-wall L-DED AISI 316L stainless steel specimens. Here, ML and PCE are used to match pointwise data from full-field DIC displacement fields (Section 3.1) with numerical FE models. This ensures that the models behave both locally and globally as the specimens and allows extracting the material properties. The material properties that are considered are the mandatory ones for a steady-state thermo-mechanical simulation, such as the thermal expansion coefficient. Detailed information considering the modeling process, the development of the metamodels, and, finally, the evaluation of the thermal expansion coefficient is presented in Section 4.1.

The part associated with predictions focuses on including the measured thermal expansion coefficient in stochastic thermo-mechanical FE models of an IHD test. The IHD test is performed using the residual strain analyzer (ReSA) experimental setup which is used for the first time for L-DED AISI 316L stainless steel parts. The ReSA experimental setup allows in situ measurements using a 3D micro-DIC system, the full-field data of which are used to compare with the results of the stochastic PCE-based metamodels. Using this approach, the displacement field around the drilling hole is predicted with good accuracy. The stochastic modeling of the displacement field for the IHD test is presented in Section 4.2.

## 3. Materials and Methods

As explained in Section 2, two tests are considered in this study. Firstly, a test to measure the thermal expansion coefficient. Secondly, an IHD test to compare with the predictions of a stochastic FE model. This section presents the specimens used for both tests and, additionally, the testing process. Note, that all specimens were manufactured using the Micron precision Milling Closed-Loop Additive (MiCLAD) machine at the Vrije Universiteit Brussel by processing PowderRange 316L powder of Carpenter Additive [16] using L-DED.

### 3.1. Thermal Expansion Coefficient

The deformation (due to thermal effects) of a solid part is related to the temperature difference applied to its volume via the thermal expansion coefficient [7,17]. Therefore, measuring the thermal expansion coefficient can be achieved in principle for any geometry if the temperature difference and the deformation due to thermal effects is known. Here, the data from Hinderdael et al. [18] are used because they offer information both for the deformation and the temperature difference.

A thin-wall specimen was chosen in the study of Hinderdael et al. [18]. The specimen was printed using 100 layers (pre-wall) and it was then stress relieved in the oven at 675 °C for 2 h 50 and air-cooled until it reached the ambient temperature. A total of 20 extra layers were later added (post-wall) on the top surface. Heat due to the addition of the 20 extra layers was transferred to the pre-wall and this resulted in changing its temperature and allowed the pre-wall to deform.

This deformation was measured by a DIC system. The DIC system measured the displacement between the initial stage where the pre-wall was cooled to room temperature and the final stage where the specimen was deformed after the addition of the 20 layers. For the DIC measurement, the same region of interest (ROI) for the initial and the final stage was considered.

In this study, the thermal expansion coefficient was determined by analyzing the deformation of the specimens that resulted from the application of heat. Additionally, the deformation of specimens under IHD was measured. DIC was employed as the method for measuring these displacements as it offers high-resolution images of the specimen’s surface and allows for highly accurate measurements of displacements, providing a comprehensive view of the deformation, as opposed to localized measurements.

Note here that the use of DIC for measuring surface displacements necessitates the use of thin-walled samples. This is because DIC is a surface measurement technique and the use of thin samples limits measurement errors that may arise from volume effects. Furthermore, thin samples tend to exhibit higher and more distinct deformations, which facilitates accurate measurement and analysis of their displacement fields using DIC.

No speckle pattern was applied, which is an uncommon practice when using the DIC technique. The authors used the roughness of the surface of the walls to reflect the incident light waves coming from an OSRAM Blue-X-Focus LED light. This allowed for in situ application of the DIC technique. Images were captured using two PointGrey cameras equipped with Edmund Optics 35 mm, f1.65 fixed focal-length lenses. The post-processing of the images was performed using the VIC-3D software version 9 by Correlated Solutions with a subset of 135 pixels and a step-size of 7 pixels. In this manner, a pixel size of roughly 15 μm was achieved, which is sufficient to accurately measure the maximum displacement values. Two images, one for the pre-wall and one for the post-wall geometry, were considered because noise effects were determined to be low [18].

Figure 1 presents the pre-wall (a) and the post-wall (b) geometries for which the contour is highlighted in blue and yellow, respectively, to aid the visualization of the borders. Furthermore, the ROI of the DIC is depicted in Figure 1c,d. In particular, Figure 1c illustrates the ROI for the reference state taken as the first image of the pre-wall after cooling. Figure 1d illustrates the ROI for the deformed state of the post-wall in a qualitative manner. The boundaries of the ROI remain unmodified for the pre- and the post-wall. The reader is referred to [18] for the printer settings and test setup of the thin walls.

Considering the above, because information of the deformation and the temperature exists for the initial and the final stage, the full-field displacement data of the DIC can be used to estimate the thermal expansion coefficient of the thin-wall specimens. However, fully exploiting the full-field data requires the experimental process to be integrated with a numerical model. The modeling of the thin-wall geometry is presented in Section 4.1.

### 3.2. Incremental Hole Drilling

Three cubes were printed for the IHD with the following MiCLAD printer settings. The laser power of 700 W at the level of the print bed was gradually decreased over the first nine layers to 450 W and kept constant then. The scanning speed was 900 mm/min, the powder feed rate 6 g/min, the shielding gas flow rate 5 L/min, and the carrier gas flow rate 8 L/min. A laser spot diameter of 1.6 mm and an overlap of 0.8 mm was used. The layer thickness, the thickness of each layer of material that is deposited onto the substrate during L-DED process, was set to 0.39 mm. A horizontal scanning direction with a bi-directional deposition pattern [5] but with a contour printed first and then a $90^{\circ }$ rotation every layer was applied. After milling the cubes and polishing the top surface, specimens with width of 12.2 mm, length of 24.9 mm, and height of 20.4 mm were obtained. A speckle pattern was applied using an airbrush on the top polished surface with a velvet white base paint. The specimens were left to dry for 24 h prior to testing. All printing parameters were chosen based on the quality of samples printed in previous studies [18].

The ReSA, shown in Figure 2, was used for the IHD measurements. It consists of a 3D micro-DIC sensor, developed by isi-sys, with a field of view (FOV) of 8.4 mm × 7.0 mm (1:1 magnification) based on Sony IMX250 2/3” CMOS, mono, 2448 × 2048 pixels (Px) @ 75 frames per second (FPS), and a Blue-X-Focus, LED light source of 200 W @ 100 Hz and 1 μs flash period. The drilling was performed in steps of 0.2 mm using a brushless air-cooled high-speed drill with torque control up to 50 k rpm and a drill hole diameter of 3.2 mm. The processing of the images of the three-dimensional measurements was performed using the VIC-3D software with a subset of 39 pixels, a step-size of 5 pixels, and a Lagrange filter for image smoothing.

Note that in experiments like the IHD where the expected strains are quite low, the noise can have a high impact on the uncertainties of the measurements. An averaging method was used to avoid excessive noise because the maximum achieved resolution corresponded to a pixel size of approximately 1 μm which is comparable to the maximum magnitude of measurements. A total of 20 images were taken at each drilling step and their mean value was kept. The mean value was used to extract the displacement field.

The results of the displacements W (along the Z direction) and U (along the X direction) for the IHD measurements are presented in Figure 3. The Z and X directions correspond to the long and short sides of the upper surface of the specimen where the speckle pattern was applied (Figure 2). The measurements of one specimen only are demonstrated because the other two gave similar results. Four circles (C0, C1, C2, and C3) are chosen near the four extremities of the drilling hole (west, east, north, south), and the average W and U displacement of the pixels inside the four circles are plotted to minimize the boundary noise. The plots are given in Figure 3a,b for drilling depths up to 3 mm and demonstrate an overall symmetric behavior. In particular, the results of the circles positioned on the Z axis (C0 and C1) present almost zero U displacement and symmetric W displacement. The results of the circles positioned on the X axis (C2 and C3) demonstrate a symmetric U displacement and almost zero W displacement.

Figure 3c,d additionally depict the contours of the displacement fields for a drilling depth of 1.2 mm. Here, the symmetric behavior is evident for the W and U displacements. Maximum values are presented near the four extremities of the drilling hole (west, east, north, south). The location of maximum values is the same for all drilling depths. The information contained in the contour plots will be elaborated on and compared directly with FE models in the following sections.

## 4. Stochastic Modeling and Optimization

### 4.1. Evaluation of the Thermal Expansion Coefficient

The thin-wall specimens are once again taken into consideration. The aim of this modeling process is to estimate the thermal expansion coefficient. The estimation is based on an optimization scheme, the concept of which can be summarized as follows.

A numerical model that behaves exactly like a specimen has the same material properties as the specimen itself. Therefore, estimating the thermal expansion coefficient reduces to finding the numerical model that gives the same displacement field as the one measured by the DIC system when submitted to the same temperature difference.

This numerical model can be developed using the same dimensions as the thin-wall geometry for the initial stage (pre-wall). Here, the geometry is not deformed and the temperature field is uniform and equal to the room temperature. For the final stage (post-wall), the addition of 20 extra layers has resulted in the change in the temperature field of the specimen, which is now deformed. This change in temperature can be simulated as a temperature field in a steady-state thermo-mechanical analysis.

Following the above, the evaluation of the thermal expansion coefficient *a* is performed by minimization of the difference between the actual displacement measured by the DIC system and the displacement estimated by the FE model, using the Nelder–Mead algorithm [19]. The difference in displacements considers every point (defined by the position vector x) that lies on the surface of the specimen measured by the DIC system ($U^{DIC}\left(x\right)$) and the corresponding points of the FE model ($U^{FE}\left(x\right)$). The displacement field $U_{i}^{FE}$ is a function of the thermal expansion coefficient $U^{FE}\left(x,a\right)$. Therefore, the objective function (function to be minimized) can be written as the root mean square error of the two displacement fields as:(1)$$F\left(a\right)=\sqrt{\frac{1}{n}\sum_{i}^{n}{\left(U^{DIC}\left(x_{i}\right)-U^{FE}\left(x_{i},a\right)\right)}^{2}}$$
where *n* is the total number of grid points. The root mean square error is a commonly employed metric of accuracy in forecast procedures and expresses the difference between the sample population values predicted by the model and the observed values [20], which makes it a perfect candidate for this specific case.

Unfortunately, the minimization approaches that are traditionally used, such as the Nelder–Mead algorithm, require the successive solution of the model until a convergence point is found. This process can be very time-consuming or even impossible considering the computational effort associated with the solution of the FE model. A metamodel is employed here as a substitute of the numerical model to reduce the computational effort.

#### 4.1.1. Development of the Metamodel

The metamodel (${\hat{U}}^{FE}\left(x,a\right)$) is a proxy model used to predict the displacement field in place of a FE model. Note that every quantity Φ predicted by a metamodel is denoted hereafter using $\hat{\Phi }$.

Two approaches are considered and compared for the generation of the metamodel. The first approach considers supervised ML [21], i.e., fitting a linear model to the displacement field evaluated by the FE model on each of the generated samples. The second approach considers the PCE-based point collocation (PC) method [22].

Note that neural networks (NN) can also be used to create metamodels. However, simple linear problems are better solved by using linear and logistic regression in machine learning, as these algorithms have a simpler structure and parameters that are easier to interpret compared to NN. In contrast to regressional machine learning algorithms, NN require a significant amount of fine-tuning of hyperparameters, which can be a complex and time-consuming task that may lead to suboptimal performance or overfitting.

Metamodels are introduced as an intermediate step between the fitting of the displacements at each point x of the surface of the sample and their respective behavior for each different value of the thermal expansion coefficient *a*. Level 0 (L0 or interior level) refers to the metamodel used to express the surface displacement field. L0 is denoted by ${\hat{U}}_{0}\left(x\right)$ and it is not a function of *a*. Level 1 (L1 or exterior level) refers to the metamodel used to express the surface displacement field ${\hat{U}}_{0}\left(x\right)$ for the different values of the thermal expansion coefficient. L1 is denoted by ${\hat{U}}_{1}\left(x,a\right)$ and it is a function of *a*.

L0 metamodels are not built on stochastic parameters because the influence of the thermal expansion coefficient is not considered. Therefore, an ML approach is used and a third-order polynomial is chosen and fitted using the least-squares method. The L0 metamodel is then formed using the fitted coefficients of the polynomial. The development of L1 metamodels can be achieved using both the PC method and the ML approach due to the stochastic nature of the thermal expansion coefficient. The two approaches are briefly outlined here.

The implementation of the PC method necessitates generating samples for the different values of the domain of interest [23] (here the different values of the thermal expansion coefficient *a*). To this end, the Sobol low-discrepancy sequence is used to generate quasi-random samples of the thermal expansion coefficient and the FE model (presented in Section 4.1.2) is evaluated on these samples, to obtain the displacement field for each sample of the thermal expansion coefficient. These sample points are stored and later used for the ML approach. A third-order polynomial expansion is generated using the three-terms recurrence method by the Stieltjes and Golub–Welsch scheme [24,25]. Finally, the metamodel is constructed as the solution to the linear regression problem defined by the displacements evaluated by the FE model (dependent variables) and the evaluations of the polynomials at the samples (independent variables). The latter is a fitting process of the coefficients of the generated expansions and it is performed using the least-squares method [26].

The coefficient of determination $R^{2}$ [22] is used for both approaches as an indication of the metamodels’ ability to capture the variability of the FE model. The coefficient for both approaches (and both levels L0 and L1) is calculated to be between 0.97 and 0.999, therefore confirming that the displacement field is accurately represented by the metamodel.

Both approaches described above are implemented using the chaospy [27] and scikit-learn [21] libraries in python, and the reader is referred to the corresponding documentation for more information.

#### 4.1.2. Numerical Modeling of Thin-Wall Specimens

The core of the present analysis is the development of FE models that accurately represent the thin-wall specimens. Here, the FE modeling is conducted using the ABAQUS software version 2022 [28] due to its versatility in handling a wide range of material models and boundary conditions, as well as its capability for post-processing and visualization of results. The FE model is built to represent a thin-wall geometry with dimensions as illustrated in Figure 4a, taken from the actual geometry of the L-DED-processed AISI 316L stainless steel (Section 3).

Due to the symmetry of the problem in planes X-Y and Y-Z, only one-quarter of the specimen is considered (Figure 4b) and symmetry boundary conditions are imposed on the respective faces. The symmetry boundary conditions in plane Y-Z restrict the U displacement and the rotations around axes Y and Z, whereas the ones in plane X-Y restrict the W displacement and the rotations around axes X and Y. The geometry is partitioned in sections along the Y axis, expressing the different printing layers that are formed during the deposition process (Figure 4b). The material used for this study is AISI 316L stainless steel with Young’s modulus 138 GPa and Poisson’s ratio 0.29. The thermal expansion coefficient, being a stochastic variable, is chosen from a uniform distribution with lower limit equal to the one of conventional steel (10.5 $\times \;10^{-6}{/}^{\circ }$C) [29] and upper limit the one corresponding to L-DED-processed AISI 316L stainless steel (17.62 $\times \;10^{-6}{/}^{\circ }$C) [30]. The bottom layer (green) is constructed using the same elastic properties but very low thermal expansion coefficient, resulting in a ‘stiff’ (with respect to temperature changes) material. This is done in order to describe the inability of the material to move due to the interaction with the print bed and the diffusion of heat due to the print bed. Both materials are visualized in Figure 4c.

The temperature field is imposed as linear between the printing plate at Y = 0 and 100th layer, following the findings in [18]. The printing plate is considered to be at ambient temperature (298 K) and the 100th layer in the temperature measured in [18]. The temperature of the 100th layer is measured in [18] to be between 700 and 1100 K and it is considered here to be a stochastic variable (similar to the thermal expansion coefficient) and follows a uniform distribution. In total, 11,600 3D quadratic reduced integration 20 node elements (C3D20R) are used, as deduced by mesh convergence studies. The meshed geometry is illustrated in Figure 4d. For the post-processing, the displacements of the nodes located on the front face (Figure 4e) are exported using a custom python script.

#### 4.1.3. Results

Based on the above, the estimation of the thermal expansion coefficient is now performed by minimizing the objective function:(2)$$F\left(a\right)=\sqrt{\frac{1}{n}\sum_{i}^{n}{\left({\hat{U}}_{0}^{DIC}\left(x_{i}\right)-{\hat{U}}_{1}^{FE}\left(x_{i},a\right)\right)}^{2}}$$
where *n* is the total number of grid points, and ${\hat{U}}_{0}^{DIC}\left(x\right)$ is an L0 metamodel fitted on the DIC data which enables the evaluation of the displacement field on the exact locations of ${\hat{U}}_{1}^{FE}\left(x,a\right)$.

The values of the thermal expansion coefficient are summarized in Table 1 for directions Z ($a_{zz}$) and Y ($a_{yy}$) for both metamodel approaches (ML and PC). The analysis of variance (ANOVA) test is used to examine if the means of the four measurements differ significantly. The results indicate that there is no significant difference between the four values at a 95% confidence level ($p_{value}=0.20$). Therefore, the findings suggest that both the ML and the PC L1 metamodels provide a similar accuracy for the measurements of the thermal expansion coefficient and along both directions. These values will now be included in the stochastic modeling of the IHD test.

Note, here, that the estimated values of the thermal expansion coefficient are close to 17.62 $\times \;10^{-6}{/}^{\circ }$C, the value that was previously reported in [30].

### 4.2. Stochastic Modeling of the Displacement Field for Incremental Hole Drilling

This section describes the stochastic modeling of the displacement field for the IHD test of an L-DED-processed AISI 316L stainless steel part.

#### 4.2.1. Numerical Modeling of the Incremental Hole Drilling

The experiment of the IHD test is performed in stages. The DIC setup is fixed and an initial reference image of the top surface is taken before the drilling. Then, the drilling is performed in stages for regular intervals of the drilling depth. Between each interval, images of the top surface are captured and compared to the initial reference image to extract the displacement field.

The displacement field is created due to the residual stresses that exist in the specimen because of the printing process. After cooling, the specimen is at a ‘pre-stressed’ state. This pre-stressed state is the one that must be taken as the reference state, before the IHD. Imposing the exact residual stress on each element in the model is impossible because it is unknown. However, an approximation can be given using models that simulate the manufacturing of the specimen, the printing, and the cooling.

In view of the above and considering that the cause of the residual stresses is the cooling of the specimen, an alternative to the modeling of the manufacturing can be achieved by the imposition of a negative steady-state temperature field. The negative temperature field simulates the cooling process, at the end of which the specimen is at the pre-stressed state with residual stresses acting in its volume. This state can then be used as a reference for measuring the displacement field. This process is simulated here using the FE method.

The FE model used to describe the IHD test is similar to the one used for the evaluation of the thermal expansion coefficient of Section 4.1.2. The geometry is presented in Figure 5a. The structure is partitioned along the X-Y and the Y-Z planes to reflect the symmetry of the problem (Figure 5b). The structure is additionally partitioned in sections along the Y axis, the drilling direction, expressing the different drilling depths. A cylinder is partitioned at the origin of the coordinate system and extended through all the layers of the specimen. This partition will be used to model the IHD test and its use is explained in the next paragraph. A linear temperature field is applied on all the layers, with 298 K at the bottom layer and the maximum temperature at the top layer. The values of the maximum temperature are considered to be uniform in the range 1050–1450 K, as measured in [18]. The temperature difference is added as negative to simulate shrinkage and induce residual stresses. The material properties coincide with those from Section 4.1.2 and the thermal expansion coefficient is sampled from the results of Table 1. In total, almost 82000 C3D20R elements are used for the simulation, as deduced by mesh convergence studies. Figure 5d presents the model in a meshed state.

Until now, a cubic specimen with residual stresses (the pre-stressed geometry) is modeled and the next step is to perform (virtually) the IHD test on this geometry. The IHD is modeled by successive simulations with different drilling depths. All the cylindrical partitions until the drilling depth are modeled using a material with very low elastic properties, thus simulating a drilled void section. An example of the geometry for different drilling depths is depicted in Figure 6. The area in red denotes the part of the specimen that is drilled and has elements with a very low stiffness. Increasing the drilling depth in successive simulations allows to simulate the IHD test.

Finally, all the displacements of the nodes of the upper surface that are not included in the cylindrical section (Figure 5e) are exported using a python script. Then, the displacement field is obtained from the difference between the reference simulation with a drilling depth of 0.0 mm and the subsequent drilling steps. This process is identical to the one used for the DIC measurements to evaluate the displacement field. The PC method is implemented using the methodology of Section 4.1.1 due to the stochastic nature of the thermal expansion coefficient and the maximum temperature.

#### 4.2.2. Results of the Displacement Field

The results of the displacement field along the X axis (U displacement) and along the Z axis (W displacement) of the IHD test of the DIC measurements are now compared to the expected value and the standard deviation of each metamodel in Figure 7, Figure 8, Figure 9 and Figure 10. Only one quadrant is considered due to the symmetry of the problem. The results are similar for all the drilling depths, and only the results for the drilling depths 1.2 mm and 2.0 mm are illustrated here. Note that the expected value and the standard deviation are estimated as the first moment and the square root of the second moment of the metamodel, respectively [31].

In general, the PC method captures well the displacement field visualized by the DIC, both in terms of the locations for the minimum and the maximum values and the overall behavior around the drilling hole. For all the drilling depths, the U displacements are higher along the X axis at Z = 0 and drop rapidly. Similarly, the W displacements are higher along the Z axis at X = 0. Furthermore, it can be observed that the results are almost symmetrical with respect to X = Z and that the maximum displacements increase as the drilling depth increases. Finally, the comparison between the PC method and the DIC measurements illustrates that the expected W displacement of the specimens (Figure 7 and Figure 9b) is between the expected and the expected plus one standard deviation of the PC (Figure 7 and Figure 9f). The expected U displacement of the specimens (Figure 8 and Figure 10b) is well represented by the expected one of the PC method (Figure 7 and Figure 9e).

## 5. Conclusions

This study focused on developing an approach for quantifying the effects of residual thermal stresses by integrating full-field experimental data from DIC images with numerical FE models using ML- and PCE-based metamodels. The approach was applied to L-DED-processed AISI 316L stainless steel parts and was presented both for measuring the thermal expansion coefficient of thin-wall specimens and predicting the displacement field around the drilling hole in IHD tests.

The measurement of the thermal expansion coefficient was realized using an optimization scheme. The optimization scheme was focused on minimizing the error between the results of the full-field DIC data and the ones of the ML- and PCE-based metamodels. The comparison between the two methods (ML and PCE) suggested that both are equally accurate in estimating the thermal expansion coefficient. Furthermore, the results achieved by the present method were close to others previously reported in the open literature.

The predictions of the displacement field around the drilling hole in the IHD tests were performed using the PC method. The comparison between the experimental results and the stochastic models illustrates that the displacement fields around the drilling hole can be predicted with an acceptable accuracy.

In view of the above, the use of ML- and PCE-based metamodels is encouraged for fully exploiting full-field DIC data and integrating experimental results with numerical FE models.

## Figures and Tables

**Figure 1 materials-16-01444-f001:**
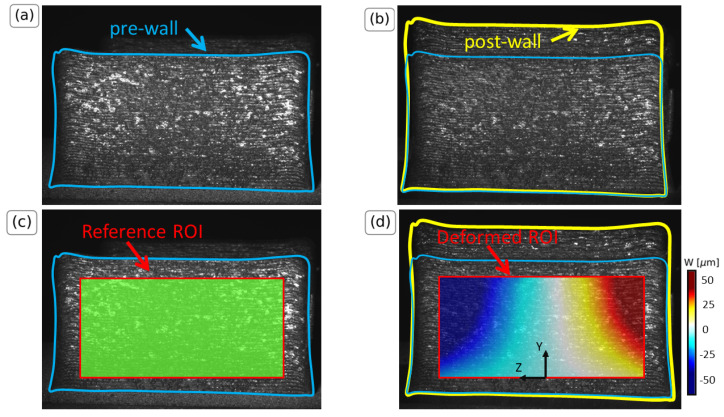
DIC images of the thin-wall samples. (**a**) Pre-wall geometry where the boundaries are indicated in blue, (**b**) post-wall geometry after the addition of 20 extra layers where the boundaries are indicated in yellow, (**c**) ROI used for the DIC for the reference state, (**d**) surface deformation W(Y,Z) of the post-wall geometry due to the expansion, where W is the in plane deformation in Z-direction. The values are indicative and are not explicitly used in the analysis.

**Figure 2 materials-16-01444-f002:**
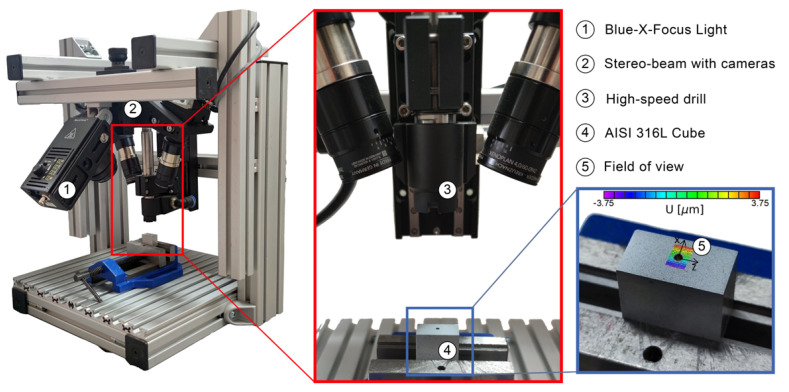
Residual strain analyzer (ReSA) experimental setup for IHD measurements.

**Figure 3 materials-16-01444-f003:**
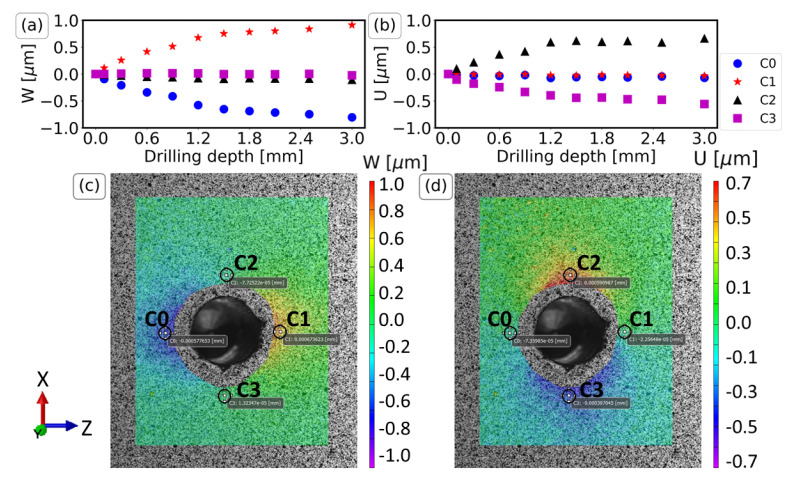
DIC results of the displacements W (**a**,**c**) and U (**b**,**d**) for the four points C0, C1, C2, C3. The plots (**a**,**b**) presents the average value for various drilling depths. The contours (**c**,**d**), as seen in VIC-3D, illustrate the displacement field for a drilling depth of 1.2 mm.

**Figure 4 materials-16-01444-f004:**
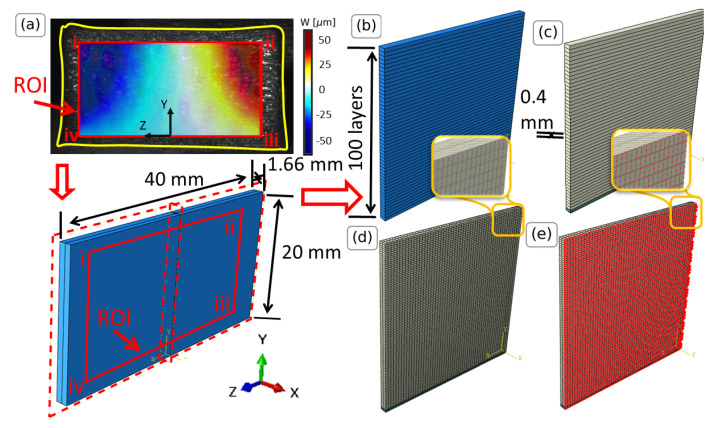
FE model of the thin-wall specimens. (**a**) Creating and partitioning of the full model due to symmetry, (**b**) thickness of printed layers, (**c**) addition of steel material (in white) and stiff base material (in green), (**d**) meshed structure, (**e**) nodes (in red) from which the displacements are extracted.

**Figure 5 materials-16-01444-f005:**
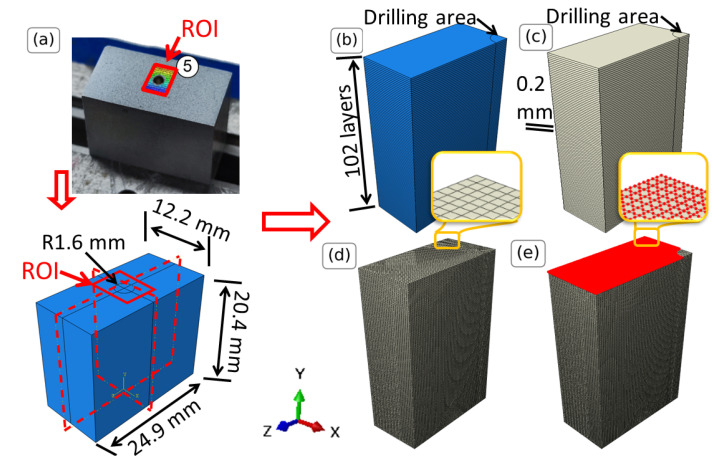
FE model of the IHD test. (**a**) Creating and partitioning of the full model due to symmetry, (**b**) thickness of printed layers, (**c**) steel material used in the modeling (in white), (**d**) meshed structure, (**e**) nodes (in red) from which the displacements are extracted.

**Figure 6 materials-16-01444-f006:**
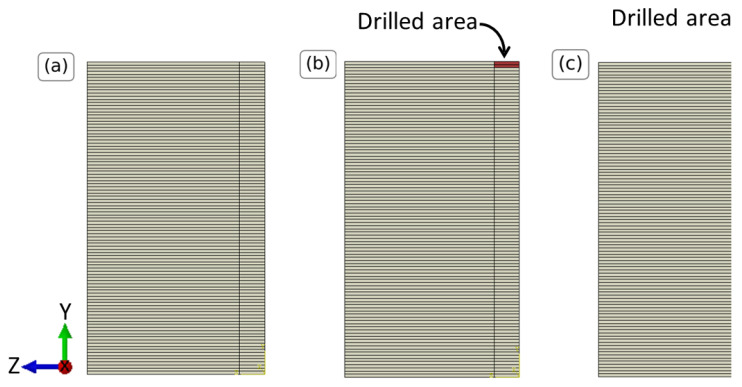
Y-Z plane view of the IHD test model illustrating the steel material (in white) and the void section (in red) for different drilling depths. (**a**) Drilling depth 0.0 mm, (**b**) drilling depth 0.4 mm, (**c**) drilling depth 2.0 mm.

**Figure 7 materials-16-01444-f007:**
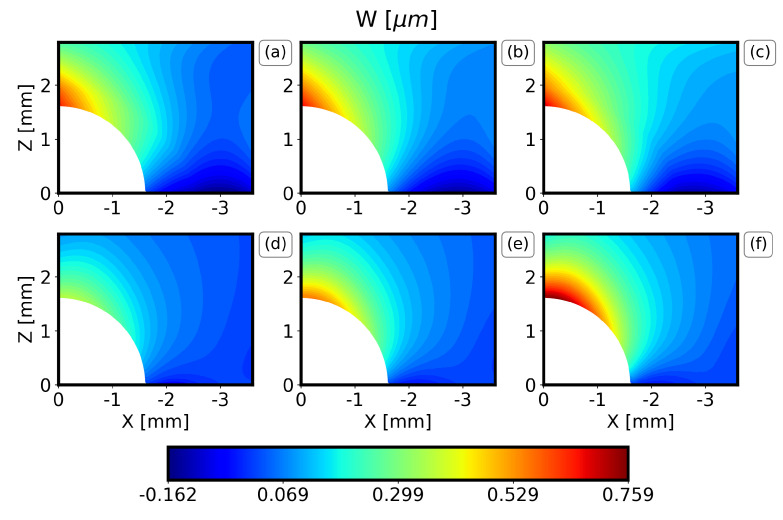
Comparison of DIC measurements and PC results for the W-displacement field and a drilling depth 1.2 mm. (**a**) Expected experimental value minus one standard deviation, (**b**) expected experimental value, (**c**) expected experimental value plus one standard deviation, (**d**) expected PC value minus one standard deviation, (**e**) expected PC value, (**f**) expected PC value plus one standard deviation.

**Figure 8 materials-16-01444-f008:**
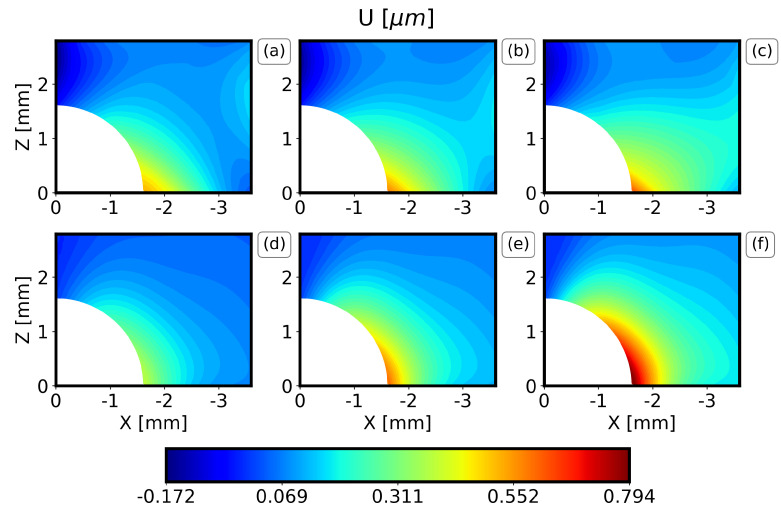
Comparison of DIC measurements and PC results for the U-displacement field and a drilling depth 1.2 mm. (**a**) Expected experimental value minus one standard deviation, (**b**) expected experimental value, (**c**) expected experimental value plus one standard deviation, (**d**) expected PC value minus one standard deviation, (**e**) expected PC value, (**f**) expected PC value plus one standard deviation.

**Figure 9 materials-16-01444-f009:**
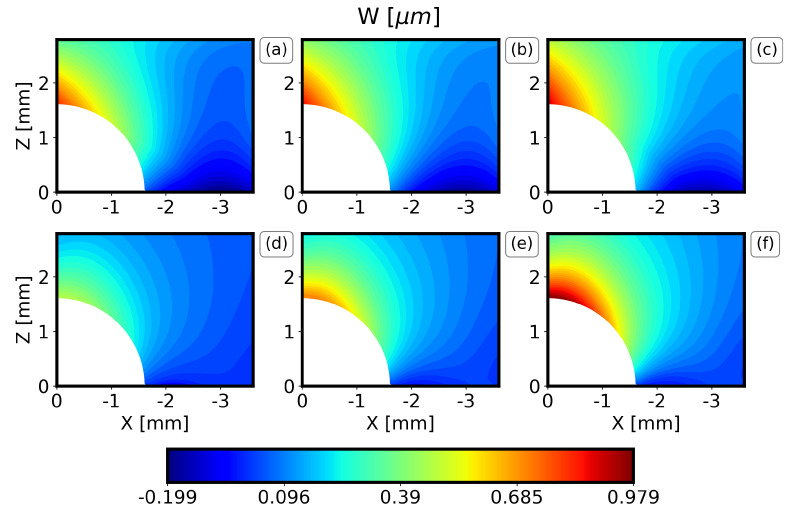
Comparison of DIC measurements and PC results for the W−displacement field and a drilling depth 2.0 mm. (**a**) Expected experimental value minus one standard deviation, (**b**) expected experimental value, (**c**) expected experimental value plus one standard deviation, (**d**) expected PC value minus one standard deviation, (**e**) expected PC value, (**f**) expected PC value plus one standard deviation.

**Figure 10 materials-16-01444-f010:**
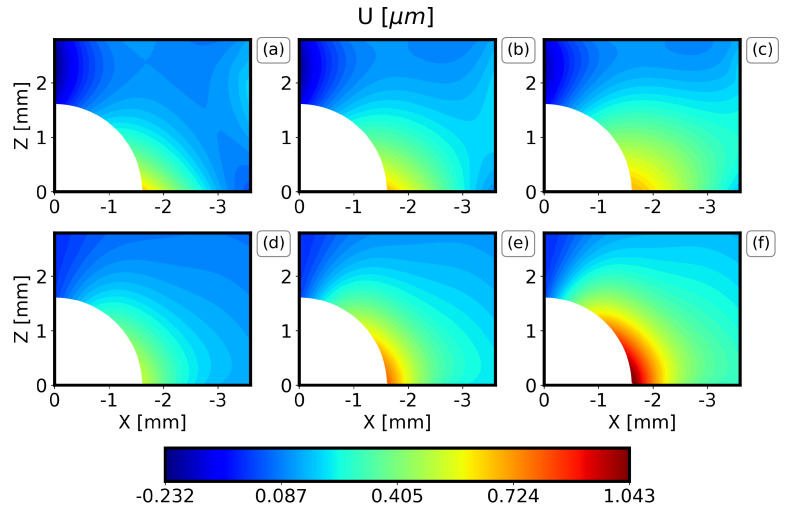
Comparison of DIC measurements and PC results for the U-displacement field and a drilling depth 2.0 mm. (**a**) Expected experimental value minus one standard deviation, (**b**) expected experimental value, (**c**) expected experimental value plus one standard deviation, (**d**) expected PC value minus one standard deviation, (**e**) expected PC value, (**f**) expected PC value plus one standard deviation.

**Table 1 materials-16-01444-t001:** Thermal expansion coefficient estimated by the minimization of the L1 metamodels (ML and PC) along the direction Z ($a_{zz}$) and Y ($a_{yy}$).

L1 Metamodel	$a_{\mathrm{zz}}$ (10${}^{-6}{/}^{\circ }$C)	$a_{\mathrm{yy}}$ (10${}^{-6}{/}^{\circ }$C)
ML	13.91 ± 2.06	17.04 ± 0.30
PC	13.76 ± 2.97	15.01 ± 1.20

## Data Availability

The research data produced in this study can be made available after contacting the authors.

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
