# Peer review of "Measuring and Predicting the Effects of Residual Stresses from Full-Field Data in Laser-Directed Energy Deposition"

_materials, 2023, doi:10.3390/ma16041444_

Round 1

Reviewer 1 Report

1. The author claimed “machine learning and polynomial chaos expansion” in the abstract section. But I did not find related contents. Can authors add more discussions?

2. Please add scale bars in Fig. 1 and 3.

3. Can authors explain “FE models”?

4. To enrich the background of laser applications, the authors should more description on laser related applications. The following references may be helpful, such as Progress in Electromagnetics Research-PIER 2023, 176, 45; Journal of Colloid and Interface Science 2023, 629, 582; Applied Physics Reviews 2015, 2, 041101.

Author Response

We would like to thank the Editor-in-Chief for the opportunity to revise the manuscript after taking into account all the valuable comments, provided by the Reviewers.

Please note that all manuscript changes have been added here in red to aid the reviewing process. All numbers of pages, lines, and references refer to the revised manuscript.

Please see the attachment for an item-by-item response to the distinct points. 

Reviewer 2 Report

It is a well-organized work and paper. Easy to follow, clearly presented in detail such that the experiments can be repeated by other researchers.

My only question is the term in the conclusion ".. results were close to reported literature values".

Can authors make a correlation between the values and put in numerical values, such as 90% correlation, 89% correlation etc?

Author Response

(The authors gave the same response as above.)

Reviewer 3 Report

The work presented in the paper is novel, But the following points need to be considered before acceptance. 

The description of figure 1 should appear below the image.

Line 109 provides an appropriate reference.

Line 114 A thin-wall specimen was chosen. Any reason?

Line 115 What is layer thickness? (Provide the information in Line 115)

Why did the authors choose DIC?  

DIC can measure load. What about load in the experiment? 

How many images have been captured?

What about the quality of the image? Why do authors not use a super image resolution system to improve image quality?'

Why do authors use 0.39 mm as layer thickness?

Why did the authors choose Abaqus?

If authors capture the images during the process, why do they not use conventional Neural Networks?

Author Response

(The authors gave the same response as above.)

Round 2

Reviewer 3 Report

The authors have incorporated the comments. Hence paper is accepted in the present from.